# Analysis of the Expansion Characteristics of Rural Settlements Based on Scale Growth Function in Himalayan Region

**Kairui Guo** 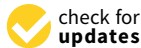**, Yong Huang * and Dan Chen**

School of Architecture and Urban Planning, Chongqing University, Chongqing 400030, China;
20141502034@cqu.edu.cn (K.G.); cd1105@cqu.edu.cn (D.C.)
* Correspondence: hyong@cqu.edu.cn

**Abstract:** Road infrastructure is reshaping the rural settlement landscape in the Himalayan area of China through the construction of the rural road and strategic highway network. However, most methods based on multiple factors described in spatial analysis of rural settlement are limited by poor spatial response mechanisms of key factors. This study provides insight into the temporal and spatial process involving 15 rural settlements of Zhada County, west of the Himalayas. The growth of rural settlement follows a "short-head S-shape" function and the general expansion rule. It indicates the mode of evolution and the characteristics of construction. The results show that 70% of rural settlements continue to report the inertia of growth, while the reconstruction of the original site leads to historical spatial displacement under spatio-temporal compression. In addition, rural settlements display a spatial organization of interface area, hinterland, and fringe area and reveal two spatial paradigms of near-road expansion and peripheral extrusion. Further, the hinterland space, which is the core of rural settlement, is compact and intensive; a quarter of the hinterland space encompasses 45% of the settlement scale. These conclusions provide guidance for delineating village boundaries and improving the human settlement environment in the Himalayan-alpine plateau.

**Keywords:** himalayan mountainous area; scale growth of rural settlements; evolution model; spatial differentiation

## 1. Introduction

The Himalayas are located on the southern edge of the Qinghai-Tibet Plateau, with an average elevation of more than 4000 m. They contain the largest glacier storage of any area except the Antarctic and Arctic [1] and are thus referred to as the Asian Water Tower [2]. However, the climate of the Himalayan-alpine plateau and the sensitive ecology lead to finite construction space and decrease agricultural production. Thus, animal grazing supported by meadow traditionally dominates the regional economy [3] and the nomadic lifestyle of rural settlements [4]. However, the nomadic method of production affects the largest grassland in the region, accounting for 70% of the regional ecosystem, and the protection is uncontrollable [5]. It seriously affects regional and global ecosystems, including water conservation, carbon sequestration, the unique biodiversity, as well as cultural heritage [6,7]. The rural settlements widely distributed across the Himalayan mountains are a key variable for ensuring water security and addressing global climate change, both of which are of great significance to human survival and stability [8]. Since 2009, to protect the plateau environment and ensure the development of the rural environment, parameters such as herdsman residency, subsidies for grassland protection, and relocation to alleviate poverty [9] have changed the scattered lifestyle into permanent and collective residence [10]. The traditional spatial organization of rural settlements in the Himalayas has undergone tremendous change. It is imperative to understand the characteristics and patterns of spatial expansion of rural settlements in the new era.

Under the new urbanization and rural revitalization scheme, reconstruction of rural physical space has long been the analytical focus in geographic, urban, and rural planning.

Rural settlement is a basic unit of rural planning and construction for the production of goods, living, and socialization [11]. Based on the theory of geospatial determinism, earlier studies focused on spatio-temporal [12] and rural landscape characteristics [13]. With the rapid influx of the rural population and resources into the urban region, the protection of the rural historical landscape [14], the urban–rural relationship [15], and rural hollowing [16] have become hotspots of research currently. The formation of rural settlements is closely related to the natural geographical environment. Therefore, most of the studies are focused on macroscale features, with geomorphological areas [17–19], watersheds [20,21], and administrative areas [22] as units. However, the rural case studies often rely on one or more specific villages [23,24]. Based on UAV mapping, 3D scanning, and other tools, it is possible to investigate the landscape composition [25] and renewal of environmental settlements [26] on a microscale. However, due to limited longitudinal data associated with finer-grain space–time resolution, most studies abstract rural settlements as construction land and residential points in order to describe the statistical characteristics and distribution status. They also ignore the inherent dynamics of rural construction or systems in the absence of responses to the spatial demands of rural residents, especially in village renewal and the construction of immigrant villages.

Road infrastructure for material and information exchange in rural areas facilitates economic and industrial transition in rural areas and directly drives spatial reconstruction of rural settlements. Especially in the Himalayas, it is also an important measure to integrate rural settlement construction [27] and ecological protection [28]. Until now, studies investigating road infrastructure and settlement have mainly included statistical analysis and recognition of driving factors. The differentiation of rural settlements is based on buffer analysis [29] and spatial overlay [30] of a geographic information system (GIS). To identify the driving factors, road infrastructure has been comprehensively analyzed based on complex factors [31], such as topography and hydrology, population economy, social culture, transportation, and public service facilities. Although, this may provide insight into the correlation between rural settlements and environmental variables, it may lead to overly generalized conclusions and logical confusion [32,33] and lack of attention to key environmental variables. From 1994 to 2013, the total mileage of roads in Tibet increased from 21,800 km to 70,000 km, and the mileage of rural roads increased 4.2-fold. In particular, the construction of the "last mile" and the strategic traffic network has changed the spatial distribution of rural settlements. Convenient transportation has lowered the cost of living, improved public services, and transformed nomadic spatial structures to road-based settlements [34]. The spatial configuration of rural settlement reconstruction under the influence of road infrastructure has basic significance for the theory and practice of the village–town system in the Himalayan-alpine plateau.

This study used two-period remote sensing data from recent decades to extract 5307 construction features of 15 rural settlements in Zhada County, Himalayan. We observed a "short-head S-shape" representing the growth of rural settlements and proposed a short-head S-shaped function to formulate variation in the rural settlement scale. It reveals the spatio-temporal response between the scale growth of inner settlement and road infrastructure. The objective of these analyses is to address the following research goals: (1) What is the interaction mechanism between rural settlement scale growth and road infrastructure in the Himalayan-alpine plateau? (2) How does road infrastructure affect the spatial pattern and construction behavior of rural settlements?

This study contributes to the literature in two ways. First, compared with studies on multi-factor-driven spatial features, this study focuses on road infrastructure, the main driving factor of rural settlement expansion in the Himalayan-alpine plateau, and proposes a "short-head S-shape" function for rural settlement scale growth. It accurately depicts the spatio-temporal evolution and spatial paradigm of rural settlements. Second, the focus on construction within the settlements bridges the gap of an internal mechanism, which is unclear in rural construction land, and residential points. It may provide a new perspective in the spatial evolution of rural settlements at high altitudes and in low density areas. These

findings facilitate the optimization of rural settlement distribution in ecologically fragile and vulnerable areas of the Himalayan-alpine plateau and support the sustainable development of human settlements. It is strategically important to implement poverty alleviation and rural revitalization measures in remote mountainous areas in the southwest China.

The next section deals with data acquisition and methods. The Results section analyzes the growth of rural settlements including the function rule, spatial patterns, and evolution characteristics in the Himalayan-alpine plateau. The last sections are Discussion and Conclusions.

## 2. Materials and Methods

### 2.1. Study Area

Zhada County is located in the westernmost part of the Himalayan mountainous area, which is a small mountain valley and basin between the northern slopes of the Himalaya and Gangdise mountain systems. It is the least populous county in China, and its residents mainly rely on agriculture and animal husbandry. Rural settlements present scattered, low-density points, so this study excludes the region's smallest villages and selects 15 central villages in Zhada County as its research object. The average altitude of these villages is above 3000 m, with one village (Riba) being found at the highest altitude of 4200 m (Figure 1). Although half of the rural settlements are township governments, the size of these settlements varies greatly due to the imbalanced population distribution and the terrain barrier in the Himalayan-alpine plateau. For example, Chusong Village, which is the government station of Chulusongjie Township, had a construction scale of only 1.5 hectares in 2018, while the construction scale of Xiangzi Village, the government station of Xiangzi Township, was already about 9 hectares in 2012.

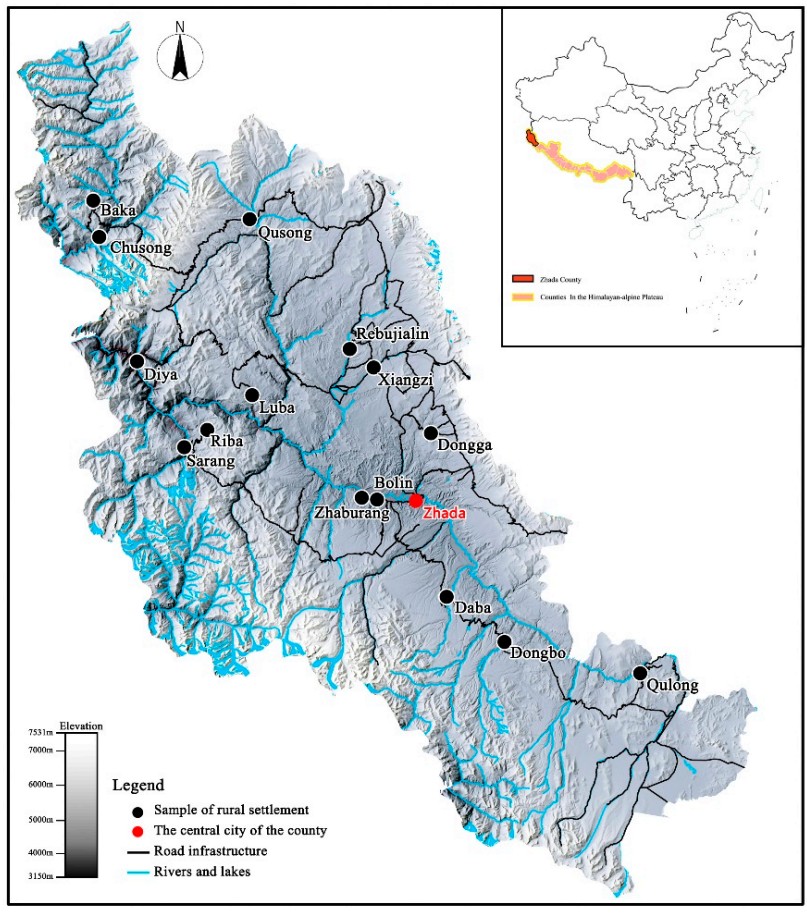

**Figure 1.** Sample distribution of 15 rural settlements in Zhada County.

*2.2. Data*

The functional landmark changes which occur in rural settlements reflect the process of settlement scale growth [35]. Common elements such as buildings, yards, and cowsheds were selected to express the material environment of rural settlements. During the investigation, we found that during the construction of economically well-off villages, these landmarks have changed dramatically. The traditional earth–rock flat buildings were replaced by brick–concrete buildings, and the space of courtyard houses and residence was divided instead of mixed. With tourism from the Qinghai-Tibet Line, photovoltaic wells and motorized farming have also promoted construction close to roads (Figures 2 and 3).

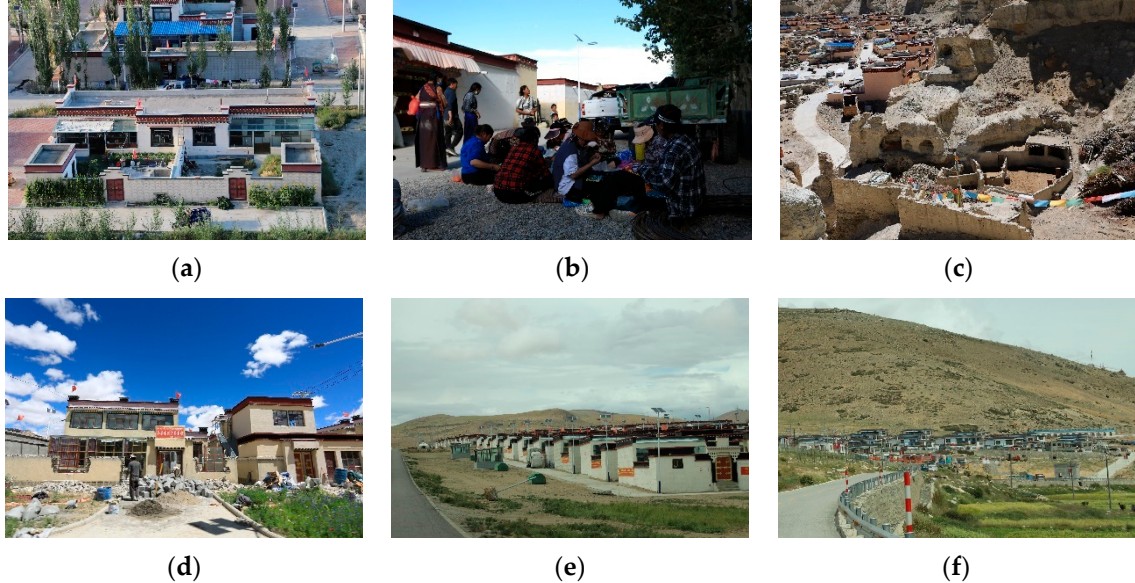

**Figure 2.** The construction of rural settlements in Himalaya (self-photographed): (**a**) the house-yard of Zhaburang; (**b**) women in Boling weaving temple supplies in yards; (**c**) old buildings turned into cowsheds in Rebugalin; (**d**) the Tibetan homestay inn in Qulong; (**e**) photovoltaic wells along the road; (**f**) motorized barley harvesting.

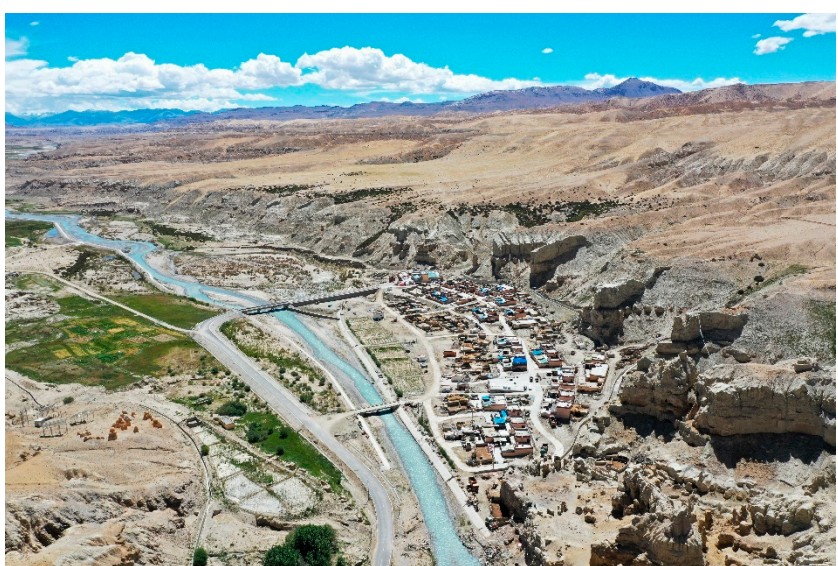

**Figure 3.** Aerial view of Rebujialin (self-photographed).

In this study, the rural construction source data came from Google historical image data, with a spatial resolution of 0.5 m. Because unified and continuous high-definition

image data could not be accessed for 15 rural settlements in Zhada County, the earliest and the latest images were taken as initial and final data, respectively (Table 1). These images were pre-processed with geometric correction, geo-registration, and size adjustment. Visual interpretation was used to identify the built-up areas, excluding isolated construction far from the continuous region, and was verified by fieldwork; 5307 patches of buildings, yards, and cowsheds were identified. Road data, defined as external connected hardened roads and excluding community roads, were based on Baidu maps and corrected using satellite image maps of the corresponding period.

**Table 1.** The image year and rural scale of 15 rural settlements.

| Name | Initial Year | Rural Scale of Initial Year (m$^2$) | Final Year | Rural Scale of Final Year (m$^2$) |
|---|---|---|---|---|
| Rebujialin | 2003 | 11,003 | 2013 | 20,507 |
| Dongga | 2005 | 10,537 | 2013 | 18,087 |
| Chusong | 2010 | 14,747 | 2018 | 15,334 |
| Riba | 2005 | 14,686 | 2013 | 20,536 |
| Luba | 2008 | 18,294 | 2013 | 20,663 |
| Baka | 2010 | 11,697 | 2018 | 17,103 |
| Diya | 2004 | 8565 | 2016 | 46,245 |
| Xiangzi | 2002 | 55,204 | 2012 | 91,949 |
| Dongbo | 2001 | 4975 | 2013 | 6261 |
| Sangrang | 2001 | 18,996 | 2013 | 25,227 |
| Zhaburang | 2002 | 38,068 | 2018 | 62,257 |
| Qulong | 2004 | 25,239 | 2011 | 34,959 |
| Qusong | 2008 | 24,260 | 2015 | 43,770 |
| Bolin | 2002 | 13,491 | 2018 | 35,279 |
| Daba | 2002 | 47,123 | 2011 | 64,531 |

*2.3. Methods*

2.3.1. Rural Settlement Scale Growth Function

Construction objects, such as buildings, yards, and cowsheds, were converted into 1 m × 1 m unit grids, and the Euclidean distance from the center of the grid to the road was calculated in ArcGis. This was done to avoid the semantic ambiguity of construction objects caused by the vector statistics of multi-layer buffers by roads in previous studies. Because the depths of rural buildings in the Himalayan-alpine plateau are about 5 m, this was taken as the distance interval used to identify the cumulative rural settlement scale change. As there was a large difference in observed scales across the 15 samples (Table 1), for the sake of comparison, the data were made into percentages from which scale cumulative scatter plots could be drawn in OriginPro 2018C (Figure 4).

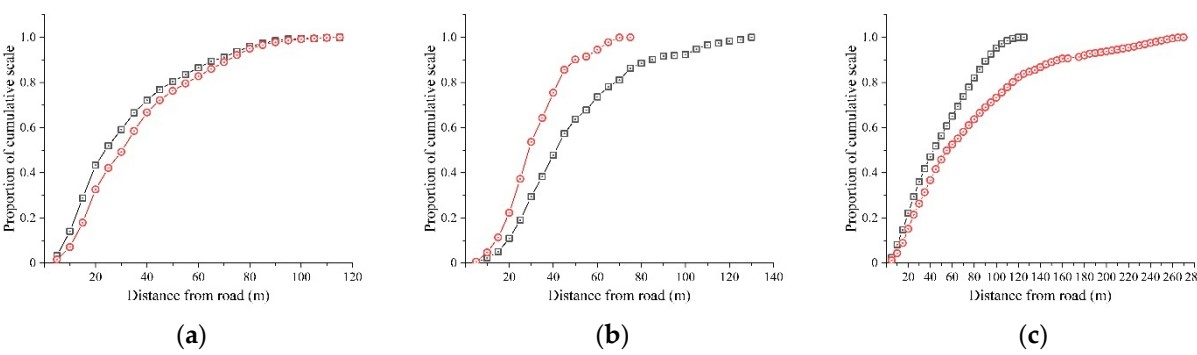

**Figure 4.** The scale cumulative map of rural settlement construction. (**a**) Chusong. (**b**) Dongbo. (**c**) Zhaburang.

The scatter plots show that the scale of rural settlement increases outwardly along the road rather than decreasing outwardly, which is in agreement with the general understand-

ing of rural settlement growth [36]. Secondly, the scale growth of rural settlements in the Himalayan-alpine plateau cannot be described by any familiar curve in the micro-scale, such as the s-shaped curves used to describe urban land density [37] and urbanization [38]. Instead, the scale growth of these rural settlements follows the "short-head S-shape" rule, representing a rapid increase near the road and little to no slow growth phase. This implies that the scale of rural settlements grows rapidly near roads, with the fastest growth being in the settlement center, with growth occurring more slowly in the fringe and periphery areas. We proposed a modified sigmoid function with an S-shape to describe the scale growth of rural settlements, which is defined as follows Equation (1):

$$f(r) = \frac{1-C}{\left(1 + e^{(r/T-1)}\right)} + D \tag{1}$$

where $f$ is the scale of the rural settlement, $r$ is the Euclidean distance from the grid to the road, $e$ is Euler's number, and $C$, $T$, and $D$ are constants. $C$ represents the growth degree of construction. $T$ represents the critical point of the concave function and convex function, at which point the scale switches from increasing to decreasing, the value is equal to the distance from the critical point to the road. $D$ is the maximum percentage of construction scale and is infinitely close to 1. The graph of this function is shown in Figure 5, with the constants $C = 2.5$, $T = 20$, and $D = 1$.

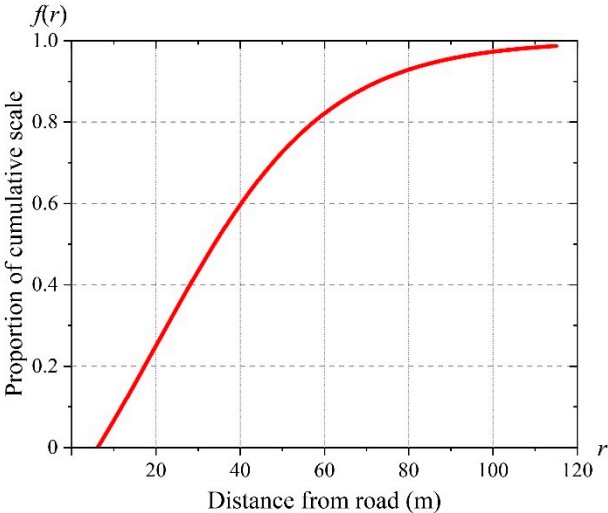

**Figure 5.** The graph of rural settlement scale growth function.

The proposed rural settlement construction scale growth function (1) is continuous, monotonically increasing, and derivable. It has two extreme points, $f = 1$ or $f = 0$. When $r = 0$ (representing the roadside), $f(r)$ equals $(1 - C)/(1 + e^{-1}) + D$, which approximates 1 because $C$ is usually about 2.5 (see the fitting result in Table 1). As r approaches infinity, $f(r)$ approaches $D$.

In this study, a non-linear least squares method was applied to fit the proposed rural settlement scale growth function from the samples. This method fits a nonlinear function to the observed data by refining the parameters in successive iterations. Trust-region algorithms, the Gauss–Newton algorithm, and the Levenberg–Marquardt algorithm are commonly used to perform non-linear least squares. The Levenberg–Marquardt algorithm was chosen and computed using OriginPro 2018C.

2.3.2. Spatial Division of Rural Settlement Scale Growth

To understand the increase of rural settlement scale from the roadside to the periphery of town, such as the rate of growth and variations on it, it is necessary to examine the derivatives of the fitted curves for rural settlement scale. This may supplement the defi-

ciencies in extant research regarding feature generalization and difference ambiguity. Let $f'(r)$ and $f'(r)$ denote the first derivative and the second derivative, respectively (Figure 6). The first derivative of the rural settlement scale growth function shows the increasing rate of the rural construction scale, and the second derivative shows the rate of the change of the increasing rate of the rural settlement scale.

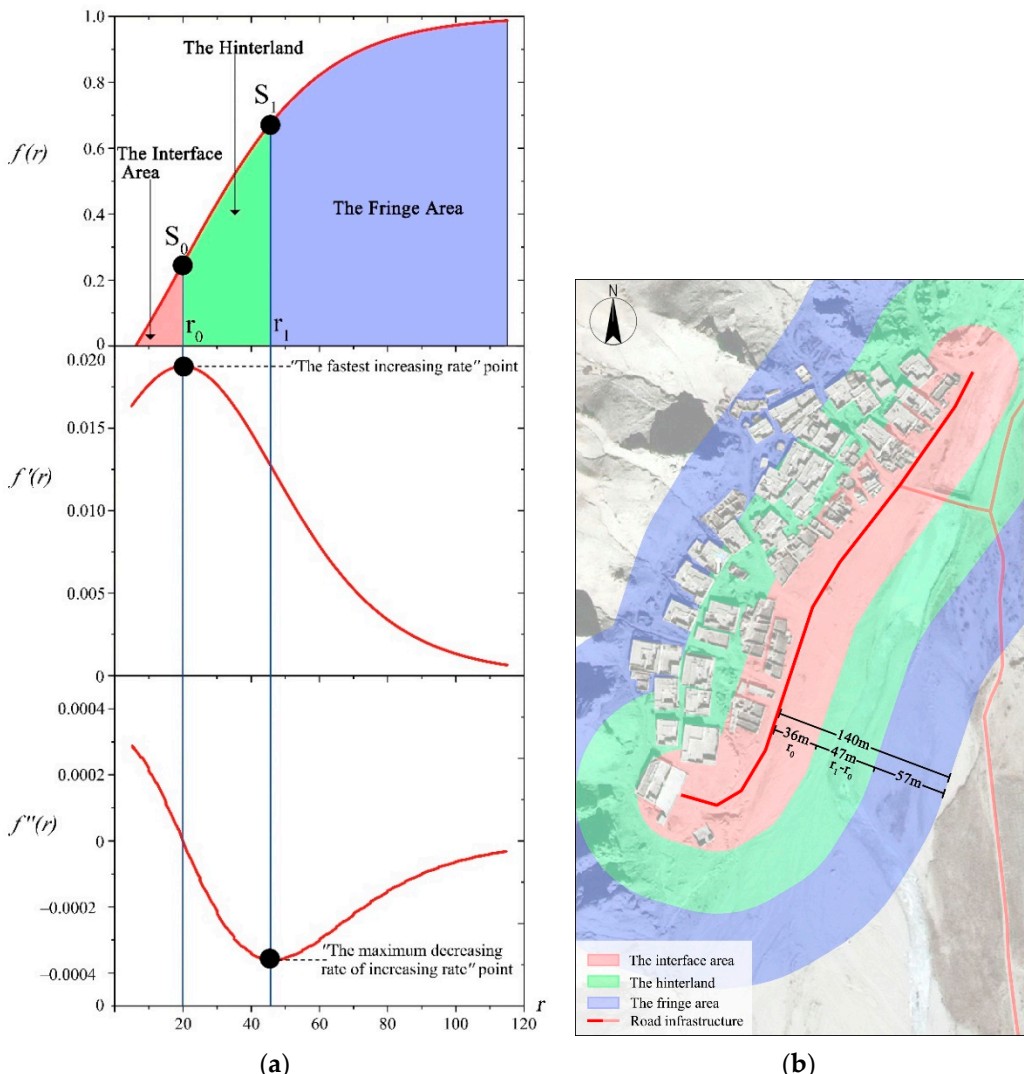

**Figure 6.** Derivative and threshold of rural settlement scale growth function. (**a**) The derivatives of scale growth function. (**b**) Spatial division of Rebujialin.

It is possible to obtain a deeper understanding of the "short-head S-shape" pattern by looking at the derivatives. From the roadside to the settlement periphery, the increasing rate of the rural settlement scale increases until it reaches the maximum and then decreases to the periphery. The changing rate of increase also changes from the roadside to the periphery. By analyzing the first derivative, the coordinates of the fastest increasing point (denoted by $S_0$ in Figure 6) can be inferred. The first derivative can be rewritten as follows (2):

$$f'(r) = \frac{(C-1)e^{\frac{r+T}{T}}}{T\left(e^{\frac{r}{T}} + e\right)^2} \tag{2}$$

Because the value of $C$ is usually about 2.5, $f'(r)$ is more than 0 and a monotone increasing function in $[0, r]$. In order to find the maximum value of $f'(r)$, the $r_0$ value

corresponding to $f'(r) = 0$, or the spatial range of maximum increment, must be examined. The second derivative of the rural settlement scale growth function is as follows (3):

$$f''(r) = \frac{(C-1)\left(e^{\frac{r}{T}} - e\right)e^{\frac{r+T}{T}}}{T^2\left(e^{\frac{r}{T}} + e\right)^3} \tag{3}$$

Let $f'(r) = 0$, so

$$\left(e^{\frac{r}{T}} - e\right) = 0$$

Thus, $r = T = r_0$, $f'(r) = \frac{1-C}{4T} = S_0$

In order to find the extreme value of the second derivative, which denotes the locations where the rates of increase and decrease change most rapidly, the $r$ value corresponding to $f'''(r) = 0$ must be examined. The third derivative of the rural settlement scale growth function is as follows (4):

$$f'''(r) = \frac{(C-1)\left(e^{\frac{r}{T}}\left(e^{\frac{r}{T}} - 4e\right) + e^2\right)e^{\frac{r+T}{T}}}{T^3\left(e^{\frac{r}{T}} + e\right)^4} \tag{4}$$

When $f'''(r) = 0$,
$\left(e^{\frac{r}{T}}\left(e^{\frac{r}{T}} - 4e\right) + e^2\right) = 0$, then $r = \ln\left(2e \pm \sqrt{3} * e\right)T$, so the $r$ value (5) and (6) can be expressed as follows:

$$r_1 = 2.317 * T \tag{5}$$

$$r_2 = -0.317 * T \tag{6}$$

The $r_2$ (6) is a negative constant, while the definition domain of the function is positive, and the negative value of the mathematical function is not considered. As a result, the maximum decreasing rate of the increasing rate of the rural settlement scale is relevant (7). So,

$$f''(r_1) = \frac{C-1}{T^3} * 3.47e^{-6} = S_1 \tag{7}$$

The rural settlement area was partitioned according to the variation of rural settlement scale growth, which is a more accurate method of partitioning based on a more intrinsic rule. Two threshold points, $r_0$ (corresponding to $S_0$) and $r_1$ (corresponding to $S_1$), could be used to clearly partition the effective area in a cumulative manner, such that three strip areas were defined from the roadside: the interface area, the hinterland, and the fringe area (see Figure 6). The interface area is closest to the road and has the fastest construction scale growth rate; the hinterland is the core area of the settlement and has a lower growth rate relative to the interface area; and the fringe area is far away from road and has the slowest growth rate.

## 3. Results

### 3.1. Fitting of Rural Settlement Scale Growth Function

The fitted curves and the estimated parameters for the rural settlement scale growth function are shown in Figure 7 and Table 2 according to the base period T value in ascending order. The average fitting precision was 0.99. The fitting function curve showed two different rural settlement scale growth processes: inertia growth and spatio-temporal displacement growth. The inertia growth was presented as an approximate overlapping curve in two periods. Although transportation is optimized in the Himalayan-alpine plateau, the growth of rural settlements still maintains the percentage of scale distribution which was present during the initial period. This reflects the increase of construction intensity with distributed inertia. About 70% of the samples were rural settlements displaying inertia growth, such as Riba and Luba (Figure 7). In comparison, spatio-temporal displacement

growth was represented by a separated curve, which shows the construction agglomeration and expansion under spatio-temporal compression. The curve of these samples changed greatly from left to right, such as in Rebujialin and Zaburangthe, where the construction range increased significantly and expanded out from the road. The growth of Dongbo, Sarang, and Qulong were agglomerated to road infrastructure.

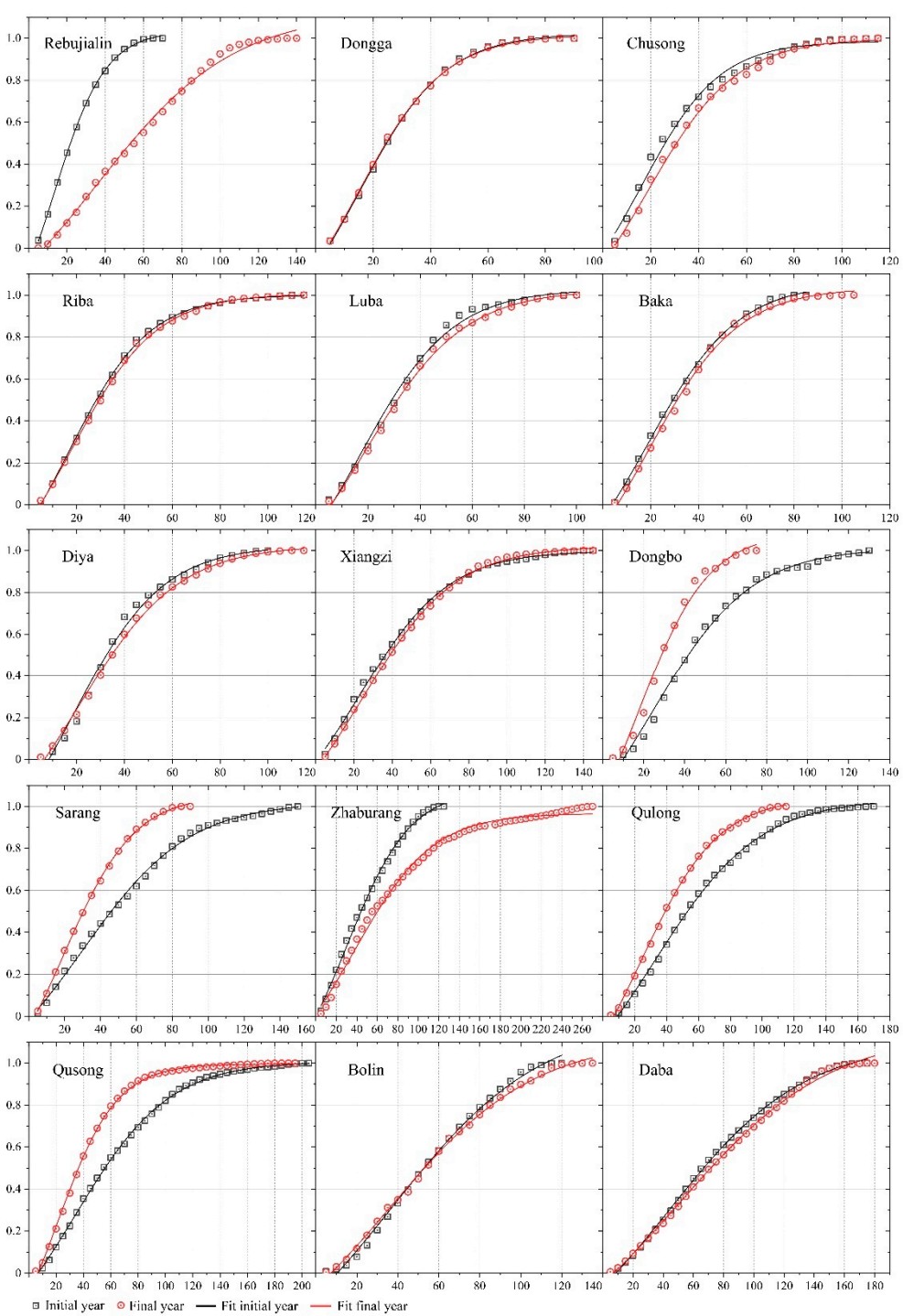

**Figure 7.** Graphs of the fitted rural settlement scale growth function.

**Table 2.** Parameters of the fitted rural settlement scale growth function.

| Name | Initial Year | | | Final Year | | |
|---|---|---|---|---|---|---|
| | C | T | $R^2$ | C | T | $R^2$ |
| Rebujialin | 2.5 | 13 | 1.000 | 2.3 | 36 | 0.996 |
| Dongga | 2.8 | 15 | 0.999 | 2.6 | 15 | 0.999 |
| Chusong | 2.6 | 16 | 0.991 | 2.5 | 19 | 0.996 |
| Riba | 2.6 | 17 | 0.999 | 2.5 | 18 | 0.999 |
| Luba | 2.5 | 17 | 0.994 | 2.5 | 19 | 0.997 |
| Baka | 2.5 | 19 | 0.999 | 2.5 | 19 | 0.997 |
| Diya | 2.5 | 19 | 0.991 | 2.6 | 21 | 0.998 |
| Xiangzi | 2.5 | 23 | 0.998 | 2.6 | 24 | 0.999 |
| Dongbo | 2.5 | 24 | 0.994 | 2.5 | 17 | 0.985 |
| Sangrang | 2.6 | 30 | 0.998 | 2.6 | 20 | 0.999 |
| Zhaburang | 2.4 | 31 | 0.997 | 2.5 | 38 | 0.994 |
| Qulong | 2.4 | 33 | 0.998 | 2.5 | 24 | 0.999 |
| Qusong | 2.4 | 34 | 0.999 | 2.5 | 21 | 0.998 |
| Bolin | 2.5 | 37 | 0.996 | 2.6 | 36 | 0.998 |
| Daba | 2.6 | 44 | 0.999 | 2.7 | 50 | 0.998 |

C: growth degree of construction, T: the spatial extent of the fastest growth of scale, $R^2$: goodness of fit.

The constant C represents the degree of rural settlement scale growth. Overall, the growth degree of most of the sample settlements improved from the initial period to the final period. However, a declining scale growth degree was observed for Zabrang village, Bolin village, and Diya village. These samples had a high proportion of demolition and construction, which was effected by the implementation of herdsman residency construction and poverty alleviation policy. The higher the proportion of demolition and construction, the faster the C value decreased. While the C value of Zabrang and Bolin fell markedly, it is important to note that both villages are newly built poverty alleviation settlements from 2018. The T value represents the spatial extent of the fastest growth of the construction scale under the influence of roads and is discussed further in Section 3.2.

*3.2. Spatial Pattern of Rural Settlement Evolution*

It can be seen clearly from the spatial partition by function derivative (Figure 8) that most rural settlements have broad fringe areas, and the proportion of marginal areas is more than 50%. The minimum range varies from 32 m in the initial period to 35 m in the final period, while the maximum range varies from 125 m to 181 m, and the average varies from 68 m to 74 m. There is a large amount of fringe variation in these rural samples. In contrast, the interface area and the hinterland are relatively compact, having a spatial range difference of between 4 m and 20 m from the initial period to the final period, with an average of 8 m. The larger spatial range in interface area and hinterland, the larger the spatial range difference between them. In the preceding decade, with road infrastructure construction in the Himalayan-alpine plateau, rural settlements formed two stable construction spaces: the interface area and the hinterland. However, the fringe area is farther from the roads and is more dependent on farmland, water, and other production behaviors, which show a great amount of variation.

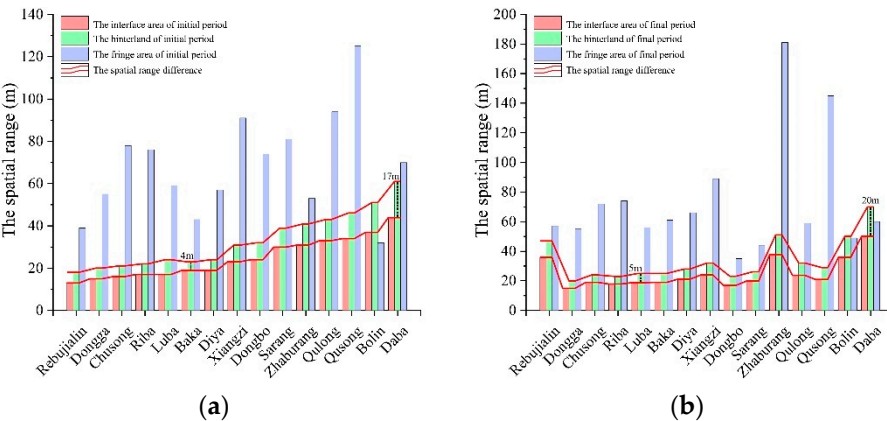

**Figure 8.** Spatial division range of rural settlement in initial year and final year. (**a**) Initial year. (**b**) Final year.

The changing proportion of the interface area and the hinterland only showed the characteristics of co-increase and co-decrease (Figure 9a). Except Dongga, all the rural settlements have changed. This change can be described by two modes: near-road expansion and peripheral extrusion (Figure 9b), without bidirectional expansion or bidirectional shrink. The expansion of the interface area drives the hinterland growth, while peripheral compression results in co-contraction of the interface area and the hinterland. As a result, the evolution of rural settlements under road infrastructure construction is not a simple expansion process but a composite space transfer.

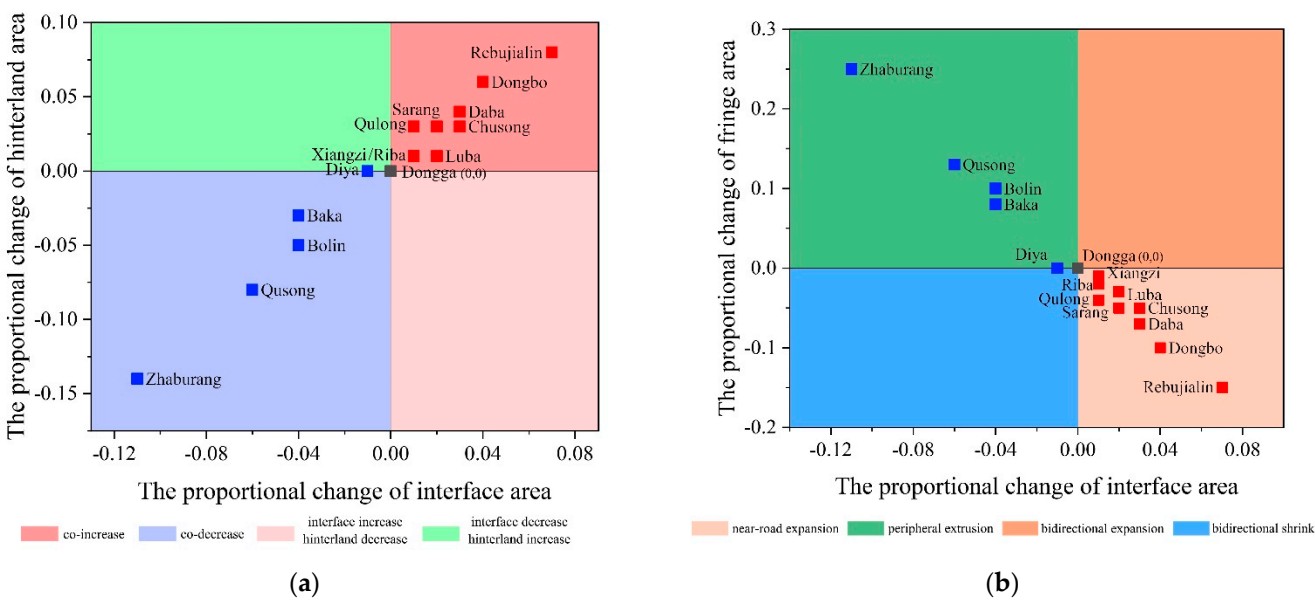

**Figure 9.** The change of spatial division proportion of rural settlement. (**a**) The interface and the hinterland. (**b**) The interface and the fringe.

### 3.3. Differentiation Characteristics of Rural Settlement Evolution

It is generally believed that the improvement of road infrastructure in plateau mountainous areas will stimulate the growth of rural settlement construction. In order to further understand how construction makes rural settlement scale growth, this study's 15 rural samples were divided into two groups based on evolutionary patterns, and the proportion of construction in three spatial divisions was examined (Figure 10). The conclusions that were drawn are as follows: (1) The distribution of the construction scale in the Himalayan-alpine plateau is approximately stable. The average proportions of construction in the

interface area, the hinterland, and the fringe area are 27%, 45%, and 28%, respectively, which reveals that in order to adapt to the plateau environment, these settlements have formed a hinterland-centered group layout. (2) The median level shows that the spatial evolution model determines the rule of distribution for the settlement construction scale. In the near-road expansion mode, the proportion of construction in the interface area increases, while construction in the hinterland and fringe areas decreases. In the peripheral extrusion model, the opposite occurs: construction in the interface area decreases, while construction in the hinterland and fringe areas increases, and the speed of construction growth decreases overall. (3) The change of box scope indicates that the floating range of the construction scale proportion is dwindling in the hinterland, and that the proportional distribution tends to be similar between settlements. However, the distribution range of the interface area and the edge area is relatively large.

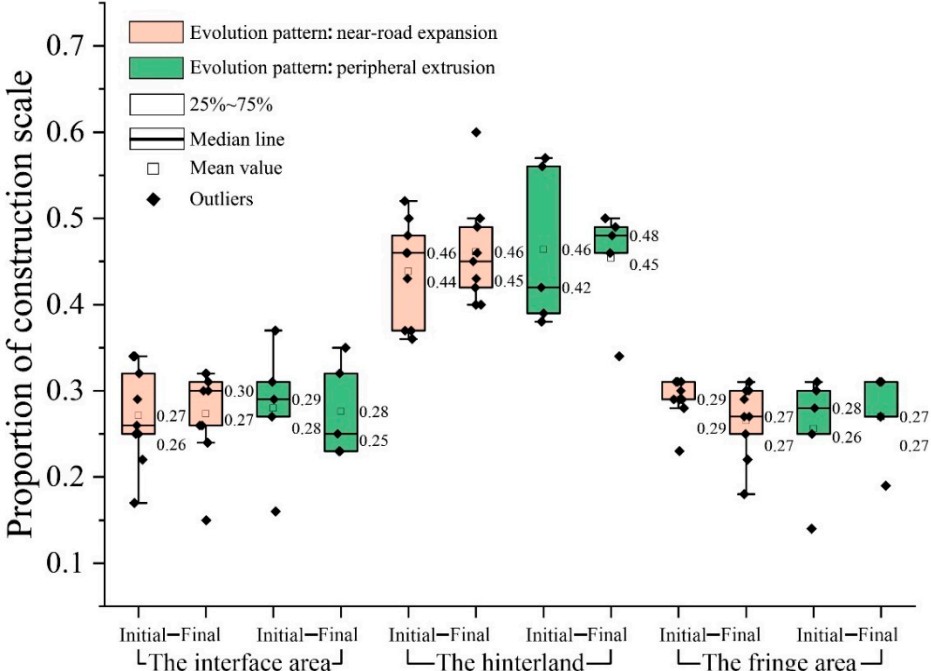

**Figure 10.** The proportion change of the construction scale in each spatial division under two expansion patterns.

It can be seen from Figure 11 that almost all rural settlements increase the proportion of buildings and dams by compressing production spaces such as enclosures. When the convenient road infrastructure makes it possible to move cattle and sheep, transport agricultural goods, and construct shelters, villagers tend to pen cattle and sheep near pastures where it is not necessary to round trips from the settlement, especially in mountainous areas where there are no grasslands nearby. Tourism from the Qinghai-Tibet line and mining also stimulate the need for the construction of buildings and yards, such as Tibetan homestay inns and dams for parking engineering vehicles. In addition, based on the proportion of different structures that have been observed in the settlements over time, the proportion of new buildings and yards is negatively correlated with the proportion of livestock enclosures, which further illustrates that an increasing amount of buildings and yards is made possible by compressing the enclosure space. Interestingly, there is a highly positive correlation between the change of yards and cowsheds in the hinterland and the change of buildings and yards in the fringe area. As a result of the new modes of life and production which are made possible by road infrastructure, the function of the hinterland infiltrates into the fringe area, which is the center of the rural settlement.

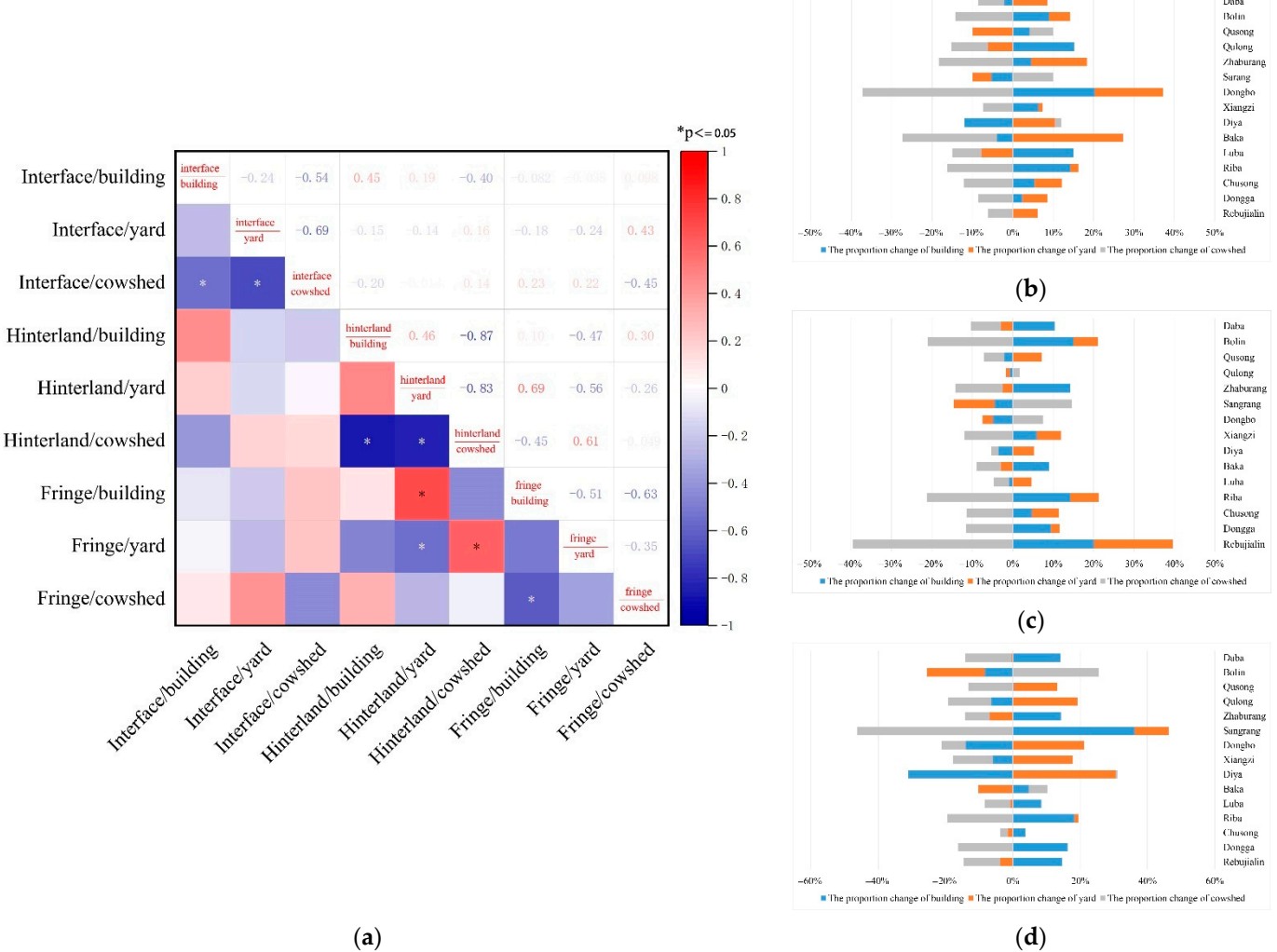

**Figure 11.** Land feature change in rural settlement construction. (**a**) The correlation of typical element proportion change in each spatial division. (**b**) The proportion change in interface area. (**c**) The proportion change in hinterland. (**d**) The proportion change in fringe area.

## 4. Discussion

### 4.1. The Dependence Path of Rural Settlement Scale Growth

The analysis of rural settlements over the past decade suggests that although the construction of all rural settlements has increased significantly, the growth of settlements with roads exhibits a "short-head S-shape" pattern. The stable relationship between environmental factors and spatial expansion has been validated in urban areas [39]. Until recently, feature description has been the normal approach to analyze rural spatial evolution. The internal mechanism underlying the growth of rural settlements scale growth has yet to be elucidated, while the study proposed a short-head S-shaped function to bridge the gap of internal mechanisms in rural settlement scale growth.

Temporally, the scale distribution of rural settlements is stable. Although the absolute scale of rural settlements has increased significantly, approximately 70% of the sample settlements have maintained the same proportional distribution that was seen in the initial period. These settlements present an inertia filling model. The growth of rural settlement scale is reflected by a near-roadway increase and outward expansion under spatio-temporal compression. However, different construction modes lead to differences in growth rules [40]. Improved road accessibility has facilitated centralization of scattered settlements in pastoral or mountainous areas [41]. Demolition and reconstruction have recombined the original

spatial texture based on an urban block, which have conformed to the parallel layout of roads for efficiency [42]. The initial growth pattern of settlement scale exhibits a downward trend, and the spatial pattern is more compact than the reconstruction mode.

The government has attempted to improve the quality of informal settlements in remote Himalayan areas such as nomadic tents and stone houses [43]. However, focusing on the efficiency of material environmental construction may neglect the ongoing historical texture inherited from the production and lifestyle in the plateau [44].

*4.2. Spatial Paradigm of Rural Settlement Scale Evolution*

The evolution of rural settlements in the Himalayan-alpine plateau presents an almost uniform spatial pattern. The spatial distribution and evolution pattern is the crux of rural settlements [45]. The fringe area of most rural settlements accounts for more than half of the space, while the interface and hinterland are relatively compact, with an average proportion of 20% to 25%. This phenomenon may be attributed to the need for irrigation beyond the peripheral farmland and water system, resulting in a broad fringe area. Spatially, not all rural settlements are the near-road expansion type. Nearly 30% of the samples expand rapidly in the fringe area away from the road, suggesting a spatial strategy to balance the job opportunities arising from road construction and the basic livelihood guaranteed by agriculture and animal husbandry in remote mountainous areas [46]. These settlements entail extensive demolition and reconstruction, which is referred to as peripheral extrusion. In this settlement type, the interface area and hinterland also increase and decrease concomitantly. This complete renewal mode focuses on the maximization of cost and physical environment, resulting in a serious impact on the historical spatial texture of rural settlements in the Himalayan-alpine plateau [47]. In the absence of adequate attention, the traditional Himalayan settlement culture will disappear in decades.

However, the observed spatial patterns and scale distribution are not uniform. Most of the construction is basically concentrated on the hinterland in these settlements. The average construction in the hinterland is approximately 45%, with a similar proportion of construction in each settlement, while the average construction in the fringe area is only 28%. Regardless of whether a settlement features near-roadway expansion or peripheral extrusion, the scale redistribution is achieved via transfer of construction from the interface to the hinterland and fringe areas. Interventions such as reduced space for livestock enclosures can be effective, allowing the settlements to increase the proportion of buildings and yards.

**5. Conclusions**

This study attempts to use functional expressions to explain the spatial rules of rural settlement expansion. It addresses the lack of universal and quantitative methods to describe the internal mechanisms of growth contributing to rural settlements based on 15 rural settlements in Zhada County, west of the Himalayas. The evolution of rural settlements away from the roadways outward exhibits "short-head S-shape" function rules. The study proposes a rural settlement scale growth function by modifying the ordinary sigmoid function. The rural settlement scale growth function for all the samples were fitted well by non-linear least squares fitting. The growth parameters of the rural settlement provide explicit physical insight into rural expansion. Based on the fitted functions, we derived a method to partition the spatial response of scale growth and road construction. The findings suggest two spatial paradigms of near-roadway expansion and peripheral extrusion, which avoid the logical confusion of multi-factor analysis reported previously. We further analyzed the differentiation characteristics of the constructed settlements to elucidate the spatial requirements. The findings support the need for spatial reconstruction of rural settlements under the ecological protection and rural revitalization program in the Himalayan-alpine plateau.

Our study illustrates the spatial progress of rural settlement growth in the Himalayan-alpine plateau and focuses on the related rules and evolution patterns observed in recent

decades. The findings indicate that the growth of rural settlements conforms to a "short-head S-shape" rule, and that 70% of villages maintained inertia during the initial period of scale growth, while others changed their traditional spatial patterns due to reconstruction. This study has also described the spatio-temporal differences resulting from the impact of road infrastructure on rural settlements. Traffic accessibility does not exclusively trigger an increase in near-road construction but serves as a growth pole centered on the hinterland. This is different from the distribution observed along the road in low-altitude mountainous areas. In recent years, the protection of plateau pastureland and the control of cattle populations led to the transformation of cowsheds into buildings and yards, which represent the internal progress of scale growth.

However, this study also has some limitations. The spatial evolution phenomenon is limited by the scale of rural settlements, which weakens the physical settlement in the spatial dimension. In addition, the accuracy of the results was not confirmed using other methods. The European spatial distance used may not address the preferences of residents for temporal variation in distance. Studies in the future should incorporate multi-model comparisons and focus on social and humanistic factors.

**Author Contributions:** Conceptualization, K.G.; Data curation, K.G.; Formal analysis, K.G.; Funding acquisition, Y.H.; Investigation, Y.H.; Resources, Y.H.; Software, D.C.; Supervision, Y.H.; Validation, D.C.; Visualization, D.C.; Writing—original draft, K.G.; Writing—review & editing, K.G. All authors have read and agreed to the published version of the manuscript.

**Funding:** This research was funded by National Key R&D Program of China (2018YFD1100804).

**Institutional Review Board Statement:** Not applicable.

**Informed Consent Statement:** Not applicable.

**Data Availability Statement:** Not applicable.

**Conflicts of Interest:** The authors declare no conflict of interest.

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
