# Peer review of "Analysis of the Expansion Characteristics of Rural Settlements Based on Scale Growth Function in Himalayan Region"

_land, doi:10.3390/land11030450_

Round 1
Reviewer 1 Report
See attached:

Reviewer 2 Report
Figure 5: Image is pixelated. Axes need to be clearly labeled in the figure, despite being explained in the text.
Line 146: “The scatter plots show…” Are these results from your research? In this section, please describe the methods and leave the results for the corresponding section of the manuscript.
Lines 228-236: This seems like methods more than results. Please move to the corresponding section.
Figure 7: Pixelated
Table 1: Abbreviations should be explained as footnotes directly on the table.
Lines 244-245: “many demolition and construction projects were observed”. This seems unsystematic.
Line 247. “newly built” When?
Figure 11: Pixelated, text on top of blue/gray background is unreadable
Line 330: “Serv-facility” is an unclear term
Discussion: This section needs an overhaul. It should discuss the results based on previous research. Only three sources are cited and in a theoretical manner. Please discuss similarities and differences compared to previous studies.
Conclusions: Recommendations or practical implications of the results should be added. Why is it useful to know that the scale growth 373 of rural settlements is subject to the 'short-head S-type' rule, and that 70% of villages 374 maintain the inertia of the initial period of scale growth, while others are changing their 375 traditional spatial patterns due to reconstruction?
Reviewer 3 Report
Dear Author(s), in my opinion the topic is very interestring and suitable with the journal, but this paper, for me, presents many major concerns.
Abstract: Is too general, i don't understand the aim of the paper and his results, is necessary to rewrite the abstract including this important informations.
Introduction: Where is the Research Question?? Which is the aim of this paper?.
Literature review: There isn't this section and in the introduction there isn't a enpugh number of previous studies able to describe the state of the art, the evolution and novelty of this paper.
Metodology: Is robust and claer, i really appreciate the immagines. But this estimates and simulation without a clear definition of the aim of this paper, they aren't enough.
Conclusion: Short and withouth scientific soundness.
Sorry for me this paper is not applicable for a publication.
Reviewer 4 Report
This paper studies the temporal and spatial process of 15 rural settlements in Zhada County, west of the Himalayas. It identifies that the scale growth of rural settlement follows the 'Short-Head S type’ function and reveals the basic mode and functional characteristics of rural settlement evolution. The following comments should be addressed carefully before further consideration.
Introduction: Some structural improvement is necessary. The introduction section should be restructured to include the following paragraphs: background, problem statement, review of relevant studies to address the problem, clear statement of the gap, research contribution(s), and the outlines of the manuscript. I could not find the research gaps and contribution statement in the introduction section.
Materials and methods:
- Figure 1: The figure is not clear. The bottom left legend is not clearly understandable. Please improve the visual quality of figure, add north sign.
- Data: this section is not well written. Please clearly mention the data used in this study, add source of data and a short but clear description regarding the particulars of the data.
- Section 2.3.1: what is “construction object”? Please add a short introduction regarding what are you going to describe?
Result: Please improve the quality of all figures.
Discussion: In this study the authors are not proposing new approach/methods. So, the practical contribution and policy implication should be elaborately discussed.
Conclusion: The conclusions section should follow this structure: summary of the problem and your contributions, the major findings in your study, the limitations and future scope of research

Round 2
Reviewer 2 Report
I believe the authors did a thorough revision and a fine job addressing the recommendations.
Author Response
Thanks so much for all the kind help to this manuscript. We have analyzed the valuable comments from the reviewers carefully, and tried our best to revise the manuscript. These comments have improved the quality of the paper immensely.
Reviewer 3 Report
I really appreciate your effort, the revised version of the paper is exhaustive and complete.
Congratulations
Author Response
非常感谢您对本手稿的所有帮助。我们仔细分析了审稿人的宝贵意见,并尽力对稿件进行修改。这些评论极大地提高了论文的质量。
Reviewer 4 Report
Although the authors improved the paper the quality of the figures is very poor. I have two issues with the figures.
a) The figures are not readable, the paper with the figures can not be published at least in the current form. The quality of the figures must be improved.
b) Even some figures are not correct. For example,
b1) Figure 8: you presented the rural settlements along x-axis. The rural settlements are discrete features (not continuos like time) with sharp boundaries. So the value can not be presented using a line graph. You must use a bar graph or any other way but not continuously.
b2) Figure 10: This figure can not be understood at all. Why the same text was duplicated? hinterland initial two times.
Similarly, you should check all the figures.
Page 15: The last paragraph was not formatted properly. Where is the conclusion section?
Author Response
Please see the attachment.

This manuscript is a resubmission of an earlier submission. The following is a list of the peer review reports and author responses from that submission.